# Study of Synthesis Pathways of the Essential Polyunsaturated Fatty Acid 20:5n-3 in the Diatom *Chaetoceros Muelleri* Using ^13^C-Isotope Labeling

**DOI:** 10.3390/biom10050797

**Published:** 2020-05-21

**Authors:** Marine Remize, Frédéric Planchon, Ai Ning Loh, Fabienne Le Grand, Antoine Bideau, Nelly Le Goic, Elodie Fleury, Philippe Miner, Rudolph Corvaisier, Aswani Volety, Philippe Soudant

**Affiliations:** 1Univ Brest, CNRS, IRD, Ifremer, UMR 6539 LEMAR, F-29280 Plouzané, France; frederic.planchon@univ-brest.fr (F.P.); fabienne.legrand@univ-brest.fr (F.L.G.); antoine.bideau@univ-brest.fr (A.B.); nelly.legoic@univ-brest.fr (N.L.G.); rudolph.corvaisier@univ-brest.fr (R.C.); 2Department of Earth and Ocean Sciences, Center for Marine Science, University of North Carolina Wilmington, 5600 Marvin K. Moss Ln, Wilmington, NC 28403, USA; lohan@uncw.edu; 3Institut Français de Recherche pour l’Exploitation de la Mer (IFREMER), Laboratoire des Sciences de l’Environnement Marin (LEMAR), UMR 6539 UBO/CNRS/IRD/IFREMER, CS 10070, 1625 Route de Sainte-Anne, 29280 Plouzane, France; elodie.fleury@ifremer.fr (E.F.); philippe.miner@ifremer.fr (P.M.); 4Department of Biology, Elon University, 50 Campus Drive, Elon, NC 27244, USA; avolety@elon.edu

**Keywords:** synthesis pathway, diatom, 20:5n-3 (EPA), *Chaetoceros muelleri*, acyl-editing mechanism, compound-specific isotope analysis

## Abstract

The present study sought to characterize the synthesis pathways producing the essential polyunsaturated fatty acid (PUFA) 20:5n-3 (EPA). For this, the incorporation of ^13^C was experimentally monitored into 10 fatty acids (FA) during the growth of the diatom *Chaetoceros muelleri* for 24 h. *Chaetoceros muelleri* preferentially and quickly incorporated ^13^C into C_18_ PUFAs such as 18:2n-6 and 18:3n-6 as well as 16:0 and 16:1n-7, which were thus highly ^13^C-enriched. During the experiment, 20:5n-3 and 16:3n-4 were among the least-enriched fatty acids. The calculation of the enrichment percentage ratio of a fatty acid B over its suspected precursor A allowed us to suggest that the diatom produced 20:5n-3 (EPA) by a combination between the n-3 (via 18:4n-3) and n-6 (via 18:3n-6 and 20:4n-6) synthesis pathways as well as the alternative ω-3 desaturase pathway (via 20:4n-6). In addition, as FA from polar lipids were generally more enriched in ^13^C than FA from neutral lipids, particularly for 18:1n-9, 18:2n-6 and 18:3n-6, the existence of acyl-editing mechanisms and connectivity between polar and neutral lipid fatty acid pools were also hypothesized. Because 16:3n-4 and 20:5n-3 presented the same concentration and enrichment dynamics, a structural and metabolic link was proposed between these two PUFAs in *C. muelleri*.

## 1. Introduction

Interest and consumption of marine products have steadily increased in recent years partly due to the health benefits provided by their high content in n-3 polyunsaturated fatty acids (PUFA) such as 20:5n-3 (eicosapentaenoic acid, EPA) and 22:6n-3 (docosahexaenoic acid, DHA) [1,2,3]. n-3 PUFAs are essential to all organisms including humans, but due to a lack of or insufficient de novo synthesis capabilities in most metazoans, n-3 PUFAs have to be obtained from the diet [4]. In the marine environment, primary producers (phototroph and heterotroph protists) are recognized for being the main suppliers of PUFAs to trophic webs [5,6,7], while on-going climate change may affect PUFAs availability and transfer across marine food webs [8,9,10,11,12,13]. Moreover, due to the high demand for human nutrition and aquaculture of carnivore species, n-3 PUFAs may become scarce because of overfishing and marine fish stock reduction [4]. Understanding how these protists synthetize EPA and DHA appears to be of a central interest prior to evaluating the influence of global changes in their availability and transfer to higher trophic levels up to human diets. Despite the economical and ecological significance of n-3 PUFAs, biological and biochemical processes responsible for the synthesis of these compounds by phytoplankton and microzooplankton are still to be fully understood.

Production of PUFAs by marine unicellular eukaryotes occurs via different metabolic pathways [14,15,16,17,18]. The most known conventional pathway is O_2_-dependent and starts with the fatty acid synthase pathway (FAS pathway), allowing fatty acid synthesis from acetyl-CoA and malonyl-CoA. It is followed by elongation and desaturation processes of the n-3 and n-6 pathways to give long-chain polyunsaturated fatty acids. These two pathways can be connected by the alternative ω-3 desaturase pathway [19]. Another route using Δ8 desaturase and bypassing the Δ6 desaturation of the n-3 and n-6 pathways has also been identified in Haptophytes microalgae such as *Isochrysis galbana* [20], *Pavlova salina* [21] or the coccolithophore *Emiliania huxleyi* [22] (Figure 1). The polyketide synthase pathway (PKS pathway) is an O_2_-independent pathway that can be found in heterotroph protists such as thraustochytrids [23,24] producing 20:5n-3 and 22:6n-3 [18,25,26].

Even if some microalgae species share all or part of these synthesis pathways, the PUFA composition of primary producers varies greatly according to species. For example, diatoms contain more 20:5n-3 than 22:6n-3, while dinophytes are richer in 22:6n-3 than 20:5n-3. As a comparison, in cyanobacteria or some chlorophytes phylum, EPA and DHA are absent or present in very low proportions (< 1% of total fatty acid (FA)) [5,6,27,28,29,30].

In higher plants as in diatoms, after synthesis by FAS pathway in the chloroplast, saturated fatty acids acyl groups have two possible destinations. The prokaryotic pathway, they can be retained in the chloroplast to form plastid glycerolipids (GL) from glycerol-3-phosphate (G3P) and serve for cell membrane structuration [31,32,33]. In the eukaryotic pathway, saturated fatty acids are transferred to the endoplasmic reticulum (ER) and acylated onto a glycerophosphate backbone to form glycerolipids and to take part in synthesis of other cell membrane lipids (such as phospholipids) or energy reserves (triacylglycerols (TAG)) [32]. According to these pathways (prokaryotic vs. eukaryotic), fatty acid composition of glycerolipids differs: GL synthesized in the chloroplast present a C_16_ acyl group at the position *sn-2*, while those produced in the ER present C_18_ or longer acyl group (C_20_, C_22_) at the *sn*-2 position [17]. In diatoms, TAG synthesis occurs in the ER from DAG [17,32]. TAG are synthesized in diatoms by the Kennedy pathway, which is acyl-CoA dependent and consists in stepwise acylations of G3P to give first lysophosphatidic acid, then phosphatidic acid, which is transformed in DAG and then in TAG by the action of acyl-CoA diacylglycerol acyltransferase (DGAT) [17,32]. Another acyl-CoA-independent pathway has been hypothesized in diatoms following the identification of enzymes and genes coding for the enzymes involved in this pathway in plants [17,34,35,36]. It would rely on the use of the phospholipid diacylglycerol acyltransferase (PDAT) to synthesize DAG from phosphatidylcholine (PC), which plays the role of an acyl donor. In plants, this pathway is supported by the PC acyl-editing machinery of the Lands cycle [32,33,37,38].

Recent studies dedicated to PUFA production have focused on studying gene expression for metabolic engineering to enhance DHA and EPA synthesis and production in microalgae or other producers initially unable to synthesize these compounds [35,39,40]. Other studies have centered on finding new genes or new enzymes implied in these FA synthesis pathways. Mülhroth et al. [35] showed that 106 genes encoding enzymes might be involved in fatty acids synthesis in *Phaeodactylum tricornutum*. However, our understanding is still incomplete especially regarding the routes followed and the production of these fatty acids of interest. Then, to improve our knowledge of PUFA production by microalgae, it is essential to better characterize quantitatively their synthesis pathways. For this purpose, ^13^C-labeling Flux Analysis offers an interesting approach that has been applied to *Escherichia coli*, yeast or microalgae [41,42,43]. The principle of stable isotope labeling relies on the monitoring of an isotopically labeled substrate incorporated into targeted organic macromolecules. Different metabolic intermediates or end products like fatty acids can be identified, and their level of labeling can be quantified, giving insights into biosynthesis pathways. Recently and thanks to instruments like Gas Chromatography coupled to Mass Spectrometry (GC–MS) or Gas Chromatography coupled to Isotopic Ratio Mass Spectrometry (GC–c-IRMS) allowing compound-specific isotope analysis (CSIA), substantial advances have been made, and it is now possible to resolve with high precision the isotopic composition of organic macromolecules including fatty acids [44,45,46,47].

In this study, a stable isotope (^13^C) labeling experiment was conducted with a monospecific culture of the diatom *Chaetoceros muelleri* in order to study the synthesis pathways of n-3 PUFA. This species has been chosen as the experimental model because diatoms are responsible for around 40% of the global primary production [48] and are characterized by a high 20:5n-3 content. The incorporation of the isotopic label (^13^C enriched CO_2_) was monitored in 10 FA, including 20:5n-3 and its potential precursors during 24 h and at a high temporal resolution (each 0.5 to 2 h). Progressive incorporation of the ^13^C into FA (from precursors to PUFA) and quantification of main fatty acid production allowed us to constrain FAS, C_16_ PUFAs pathways and n-3 and/or n-6 elongase/desaturase pathways and their involvement in EPA production by *C. muelleri*. To monitor its growth and physiological status, several cellular parameters (morphology, viability, esterase activity and lipid content) were also concomitantly measured by flow cytometry analysis (FCM).

## 2. Materials and Methods 

### 2.1. Algal Culture and Isotopic Labeling

Monospecific cultures were performed with the marine diatom *C. muelleri* (strain Culture Collection of Algae and Protozoa (CCAP) 1010-3 obtained from the CCAP culture collection of the Scottish Marine Institute, Oban, Scotland, UK, formerly listed as *Chaetoceros neogracile* VanLandingham 1968) in batch (total culture volume of 2 L) and under continuous light. Cultures were prepared with 1750 mL of sterile filtered seawater, 250 mL of algal inoculum and 2 mL of nutrients medium (Conway medium) and kept sterile during the whole experiment. Three culture balloons were prepared: two replicates for ^13^C labeling (balloon 1, Cm1 and balloon 2, Cm2) and one as a control (CmT) without ^13^C enrichment and fed with petrochemical CO_2_ (whose isotopic composition is equal to −37‰).

The three culture balloons (two labeled and one control) were subjected to a pre-culturing phase of 4 days before the start of the monitoring in order to sample the alga during the exponential growth phase and to promote label assimilation into FA. The isotopic labeling was performed using pure ^13^C-CO_2_ gas (Sigma-Aldrich, < 3%atom 18O, 99.0%atom ^13^C). The control balloon was continuously bubbled with unlabelled CO_2_ (i.e., petrochemical CO_2_). For each of the three replicates, the CO_2_ introduction in the culture was started just before the first sampling time (t_0_) and maintained during 24 h (t_24_). The growth of the two labeled replicates was controlled by ^13^CO_2_ addition using a pH-stat system, which fed the culture when pH was higher than 7.50 ± 0.05 (Figure 2). The whole experimental system was designed in order to keep the cells in a good physiological state (controlled by flow cytometry analyses) during the experiment.

### 2.2. Samples Collection

Sampling was performed during 24 h at 30 min, 1 h, 2 h, 3 h, 4 h and then every 2 h. A total of 16 samples were thus collected during the 24 h monitoring. The sampling system (Figure 2) was designed to collect the culture without opening the balloon and to avoid bacterial contamination and the introduction of atmospheric CO_2_. At each sampling time, a total volume of 52 mL per balloon was collected: (i) 2 mL for flow cytometry (FCM) analysis of cellular parameters, (ii) 25 mL for isotopic analysis of Particulate Organic Carbon (^13^C-POC) and Dissolved Inorganic Carbon (^13^C-DIC) by Elemental-Analyzer Isotope Ratio Mass Spectrometry (EA-IRMS), (iii) 25 mL for fatty acid analysis in neutral lipids (NL) and polar lipids (PL) by Gas-Chromatography Flame Ionisation Detection (GC-FID) and (iv) compound-specific isotope analysis (CSIA) of FA (^13^C-FA) as described in the following paragraphs.

### 2.3. Flow Cytometry Analysis

Algae growth cellular variables were measured using an Easy-Cyte Plus 6HT flow cytometer (Guava Merck Millipore^®^, Darmstadt, Germany) equipped with a 488 nm blue laser, detectors of forward (FSC) and side (SSC) light scatter, and three fluorescence detectors: green (525/30 nm), yellow (583/26 nm) and red (680/30 nm). Cell variables i.e., forward scatter (forward scatter, FSC), side scatter (side scatter, SSC) and red fluorescence (FL3, red emission filter long pass, 670 nm) were used to identify and select the *C. muelleri* cell population. FSC and SSC give, respectively, information on the relative size and complexity of cells [49,50,51]. The flow cytometry measurements were performed on fresh (living) samples.

Three fluorescent probes were used to assess viability, esterase activity, and lipid content according to Seoane et al. [52]. Briefly, the SYTOX (Molecular Probes, Invitrogen, Eugene OR, USA, final concentration of 0.05 µM), a probe that binds to the DNA of a permeable or permeabilized cell, was used to estimate the percentage of dead cells in culture sample [53]. The fluorescein diacetate (FDA, Molecular Probes, Invitrogen, Eugene OR, USA #F1303 at a final concentration of 6 µM, emission wavelength 525 nm), a substrate for intracellular esterases, was used to provide a proxy of primary metabolism. As cell membrane integrity is essential for the retention of FDA product (fluorescein) by the cell, it is also used to estimate the percentage of metabolically active cells [52]. Cells with a high level of green fluorescence (FL1) correspond to high esterase activity, while cells with a low FL1 fluorescence are considered cells with reduced esterase activity (reduced metabolism) or dead/dying cells. The BODIPY probe (BODIPY 505/515 FL; Molecular Probes, Invitrogen, Eugene OR, USA, final concentration of 10 mM), which stains lipid droplets within microalgae cells, was used as a proxy of lipid reserves [52]. The green fluorescence emitted is proportional to the quantity of lipid reserve present in the cells. 

The concentration of bacteria was also monitored during the experiment according to Seoane et al. [52] by using SYBR^®^Green (Molecular Probes, Invitrogen, Eugene OR, USA, #S7563), a DNA staining fluorescence probe which allows detection of DNA stained bacteria on FL1 detector (green fluorescence). Results are given as concentration of bacteria cells per mL.

### 2.4. POC Concentration and Bulk Carbon Isotopic Composition

Samples (25 mL) for POC concentration and stable isotopic composition were filtered through pre-combusted 0.7 μm nominal pore-size glass fiber filters (Whatman GF/F, Maidstone, UK). The filter was then dried at T = 55 °C, fumed with hydrochloric acid to remove particulate inorganic carbon, subsampled with a 13 mm punch and encapsulated into tin caps for further analysis ([54,55] and references therein).

POC concentrations of all samples were measured using a CE Elantech NC2100 (Thermo Scientific, Lakewood, NJ, USA) according to the protocol of the United States Environmental Protection Agency [56] (with acetanilide (99.9% purity, C_8_H_19_NO CASRN 103-84-4) as a standard. Briefly, the encapsulated samples were combusted at 980 °C in the elemental analyzer. The combustion products were passed over a copper tube with chromium oxide/cobaltous acetate oxide as catalysts to aid the conversion of carbon into carbon dioxide. The mixture was released to thermal conductivity detectors to measure the levels of carbon in a sample ([55] and references therein). The measured values are corrected with those of known standards. POC concentration is given in mmol·L^−1^. 

Bulk carbon isotopic composition (^13^C POC) was analyzed by continuous flow on an Elemental Analyzer (EA, Flash 2000; Thermo Scientific, Bremen, Germany) coupled to a Delta V+ isotope ratio mass spectrometer (Thermo Scientific, Bremen, Germany). Calibration was performed with certified international standards and in-house standards described in Table 1.

### 2.5. DIC Concentrations and Bulk Carbon Isotopic Composition

Samples for Dissolved Inorganic Carbon concentration and stable isotope composition were collected from the filtrate of POC samples (25 mL). Twelve milliliters was put into Labco Exetainer (Labco, Wales, UK) tubes, poisoned with 20 µL of mercury chloride (HgCl_2_) and stored at 4 °C until Gas-Bench Isotope Ratio Mass Spectrometer (GB–IRMS) analysis. The sample preparation for DIC concentration and ^13^C-DIC measurements was made according to Assayag et al. [57]. Briefly, 1 mL subsample was added to the mixed tube, then 23 droplets of phosphoric acid (H_3_PO_4_) were introduced into a closed Exetainer tube (Labco, Wales, UK) that is flushed with ultra-pure helium gas to eliminate the residual air and avoid contamination. The H_3_PO_4_ addition converted all DIC species into CO_2_ [57], and after 15 h, equilibration at room temperature CO_2_ was measured in the headspace of a Gas Bench coupled to a Delta Plus mass spectrometer from Thermo Scientific, Bremen, Germany (GB–IRMS).

### 2.6. Isotopic Data Processing

Because we consider here ^13^C-enriched samples, we used the atomic abundance of ^13^C in percentage (%atom of ^13^C) to express the results. Conversion between delta notation and %atom^13^C notation can be done as follows [58]:(1)%atomC13=100×(δC131000+1)×(C13C12)VPDB1+(δC131000+1)×(C13C12)VPDB

With (^13^C/^12^C)_PDB_ = 0.0112372, the ratio of ^13^C to ^12^C in the international reference Vienna-Pee Dee Belemnite (V-PDB) standard.

Atomic enrichment (AE) is then calculated from %atom^13^C POC-corrected by POC_control_ values (1.08% close of the natural values observed in the marine environment) and atom %^13^C DIC-corrected by control abundance values (DIC_control_ = 1.12%) according to the following equations:(2)AEPOC=%atomC13−POCcontrol
(3)AEDIC=%atomC13−DICcontrol

### 2.7. Fatty Acids Analysis

#### 2.7.1. Lipid Extraction

After filtration on pre-combusted 47 mm GF/F filters (porosity 0.7 µm) of 25 mL of culture, lipases were deactivated by addition of boiling water, and lipids were extracted by diving the filter into 6 mL solvent mixture (chloroform:methanol, 2/1, v/v). Lipid extracts were flushed with nitrogen and stored at −20 °C until analysis.

#### 2.7.2. Separation of Neutral and Polar Lipids

Total lipid extracts were separated into neutral and polar fractions following the method of Le Grand et al. [59]. In brief, 1 mL of extract was evaporated to dryness under nitrogen, recovered with 3 washes of 0.5 mL of chloroform:methanol (98:2 v:v; final volume 1.5 mL) and spotted at the top of a silica gel column (40 mm × 4 mm, silica gel 60A 63–200 µm rehydrated with 6% H_2_O, 70–230 mesh, Sigma-Aldrich, Darmstadt, Germany). Neutral lipids fraction (NL) was eluted using chloroform:methanol (98:2 v:v; 10 mL) and polar lipid fraction (PL) with methanol (20 mL). Both were collected in glass vials containing an internal standard (C23:0, 2.3 µg). Lipid fractions were then stored at −20 °C until further analysis.

#### 2.7.3. Transesterification of FAME

Fatty acids methyl esters (FAME) transesterification was conducted according to the protocol described by Mathieu-Resuge et al. [60]. In brief, after evaporation to dryness of the neutral and polar lipid fractions, transesterification was made by adding 0.8 mL of H_2_SO_4_/methanol mixture (3.4%, v:v) to the lipid extract and heated at 100 °C for 10 min. Hexane (0.8 mL) and distilled water saturated with hexane (1.5 mL) were added. The lower MeOH–water phase is discarded after homogenization and centrifugation. Hexane fraction containing FAME was washed two more times with another 1.5 mL of distilled water. 

#### 2.7.4. Fatty Acid Analysis by Gas Chromatography Flame Ionisation Detector (GC-FID)

Analysis of FAME was performed on a Varian CP8400 gas chromatograph (Agilent, Santa Clara CA, USA) using simultaneously two parallel columns: a polar (DB-WAX: 30 mm × 0.25 mm ID × 0.25 µm, Agilent, Santa Clara CA, USA and apolar column (DB-5: 30 m × 0.25 mm ID × 0.25 µm, Agilent, Santa Clara CA, USA). The temperature program used by the gas chromatograph was the following: first, an initial heating to 0 from 150 °C at 50 °C·min^−1^, then to 170 °C at 3.5 °C·min^−1^, to 185 °C at 1.5 °C·min^−1^, to 225 at 2.4 °C·min^−1^ and finally to 250 °C at 5.5 °C·min^−1^ and maintained for 15 min. The FAME were identified by comparison of their retention time with commercial and in-house standards mixtures as shown by the two columns used in Appendix A. 

Fatty acid concentrations are expressed in µmolC·L^−1^. Fatty acids are grouped according to their hypothesized synthesis pathways as described in Table 2.

#### 2.7.5. Fatty Acids Compound-Specific Isotope Analysis

Compound-specific isotope analyses (CSIA) of FAME were performed following the protocol described by Mathieu-Resuge et al. [60]. CSIA analyses were made on a Thermo Scientific (Bremen, Germany) GC ISOLINK TRACE ULTRA using the same apolar column as for FAME analysis by GC–FID (DB-5: 30 m × 0.25 mm ID × 0.25 µm, Agilent, Santa Clara CA, USA). Only the fatty acids with the highest concentrations and involved in the previously described pathways as measured by GC–FID analyses were considered for CSIA (concentrations superior to 250 µg·L^−1^). The minor FA presenting a too low signal amplitude (< 800 mV) on the GC-c-IRMS (generally below 100 µg·L^−1^) did not allow precise isotope ratio analysis and were not considered. Although 18:1n-9 and 18:3n-3 co-eluted on the apolar column of GC-c-IRMS, most of the isotopic signature is attributed to 18:1n-9. The amount of 18:1n-9 is about 55 times higher than the 18:3n-3 in neutral lipids (NL) and about 30 times higher in polar lipids (PL) in *C. muelleri* (49.7 ± 5.8 versus 0.9 ± 0.4 µmolC·L^−1^ for NL, respectively, for 18:1n-9 and 18:3n-3 and 24.7 ± 5.1 versus 0.9 ± 0.3 µmol·L^−1^ for PL, respectively, for 18:1n-9 and 18:3n-3).

The GC–c-IRMS measured values were calibrated using the F8-3 standard mixture of eight fatty acid ethyl and methyl esters (14:0, 16:0, 18:0 and 20:0 with values ranging from −26.98 ± 0.02‰ to −30.38 ± 0.02‰) supplied by Indiana University Stable Isotope Reference Materials (Schimmelman, Indiana University IN, USA) as described in Mathieu-Resuge et al. [60].

#### 2.7.6. Compound-Specific Isotope Data Processing

FA atomic enrichment (AE_FA_) is calculated with the same method as for AE_POC_ and AE_DIC_, with %atom^13^C for each FA given by GC-c-IRMS analysis (AE_FAcontrol_ = AE_FAnat_ = 1.08%). AE_FA_ is then used for estimating the specific uptake rate in FA (µ_FA_ in h^−1^) with the following equation: (4)$${µ}_{FA}=\frac{\frac{AE_{FA}}{AE_{DIC}}}{t_{24}-t_{6}}$$

µ_FA_ is calculated between t_6_ (once the DIC ^13^C enrichment started) and t_24_ (end of the enrichment experiment). AE_FA_ corresponds to cell enrichment at t_24_ (AE_FA_ at t_6_ is considered as negligible). The AE_DIC_ is calculated using the time-weighted average of AE_DIC_ over the same time period.

Finally, to evidence FA synthesis conversion from fatty acid A and fatty acid B in *C. muelleri*, we calculated the AE_FA_ ratio (R) of product B over supposed precursor A. If the calculated ratio is close to 1 with a confidence interval at α = 0.1, the fatty acids A and B are supposed at equilibrium and synthesized simultaneously or very closely. If the ratio is below 1, the transformation of A into B is possible but slower. On the contrary, if the ratio is above 1, A is not considered as a precursor of B. R is defined as follows.
(5)$$R\;=\frac{{\mathrm{AE}}_{\mathrm{FA}\left(B\right)}}{{\mathrm{AE}}_{\mathrm{FA}\left(A\right)}}$$
with A as the fatty acid that is supposed to be a precursor of fatty acid B and AE_FA(A)_ and AE_FA(B)_ their respective atomic enrichments at each sampling time.

### 2.8. Statistical Analysis

To assess the potential effect of time and difference between balloons during algae development and ^13^CO_2_ incorporation, PERMANOVA analysis was used on the FA mass percentage separately in NL and PL. Principal component analysis (PCA) coupled with similarity percentage analysis (SIMPER) allowed us to distinguish fatty acids that are the main ones responsible for the overall observed variability (80%). A Spearman test was also conducted on fatty acids abundance in both PL and NL to explore the relationship between fatty acids. All statistical analyses were conducted using R software.

## 3. Results

### 3.1. Algae Physiology During Growth

During the 24 h of the experiment, the replicability between balloons (enriched (Cm1 and Cm2) vs. control (CmT)) was satisfying with similar increasing trends despite slightly lower concentrations of the control ballon. *C. muelleri* grew on average from 8.3 × 10^6^ cells·mL^−1^ at t_0_ to 11.3 × 10^6^ cells·mL^−1^ at t_24_, corresponding to an increase factor of 1.4 (Figure 3). By considering the time point 0.5 (average of 6.6 × 10^6^ cells·mL^−1^), this growth was closer to a doubling of cell density (factor 1.7). After t_16_, a sharper increase was observed for the three balloons with a change in general slope (9.2 × 10^5^ cells·mL^−1^·h^−1^ to 19.1 × 10^5^ cells·mL^−1^·h^−1^). Bacteria concentrations increased from 1.1 × 10^8^ cells·mL^−1^ (t_0_) on average to 1.4 × 10^8^ cells·mL^−1^ t_24_. The strongest increase in bacteria concentrations was observed between t_0_ and t_14_. After t_14_, the bacteria content concentration reached a plateau at 1.4 × 10^8^ cells·mL^−1^ for enriched balloons and 1.2 × 10^8^ cells·mL^−1^ for the control balloon, being slightly lower than enriched balloons (factor 1.2) (Figure 3).

During the experiment, cell size (FSC), cell complexity (SSC) remained relatively constant and similar between control and enriched balloon (respectively 153.3 ± 1.7 a.u (Student test, *p*-value < 0.001 and ANOVA *p*.value = 0.3) and 52.8 ± 2.5 a.u (Student test, *p*-value < 0.001 and ANOVA *p*-value = 0.07), on average for the three balloons) (Table 3). Chlorophyll content (FL3) decreased slightly between t0.5 and t24 from 274 a.u to 257 a.u. Percentage of dead cells as measured by SYTOX DNA staining (Green fluorescence, FL1) remained below 4% and averaged around 2%, while the percentage of metabolically active cells averaged at 90% with the lowest value at 86%. It revealed that *C. muelleri* cells were in good health for the whole experiment. The neutral lipid content as estimated by BODIPY staining, (green fluorescence, FL1) was more variable. It averaged at 225.3 a.u, but time points t_0_, t_6_, t_10_, t_12_, and t_18_ were lower (166.0 ± 16.0 a.u), with the lowest value at t_0_ (154.1 a.u) and the highest at t_20_ (308.5 a.u) (Table 3).

### 3.2. Correlation between POC Concentration, TFA Concentration, and Cell Abundance

Particulate Organic Carbon concentration increased slowly between t_0_ and t_24_ for the three replicates from 12.3 ± 2.6 mmol·L^−1^ to 22.4 ± 1.6 mmol·L^−1^ (Figure 4). Total Fatty Acids (TFA) concentration increased progressively with time between 2.3 ± 0.3 mmol·L^−1^ to 4.5 ± 0.2 mmol·L^−1^ (Figure 4). POC concentration was linearly correlated to cell abundance (y = 2.4x − 5.7, R^2^ = 79%, *p*-value < 0.001) and to TFA concentration (y = 0.23x − 0.5, R^2^ = 80%, *p*-value < 0.001) (Figure 4). The slope of the first regression is 2.4 × 10^−3^ nmol.cells^−1^ which is a proxy of carbon content per cell (Figure 4). The slope of the regression between TFA and POC concentration gives an estimation of the proportion of fatty acids in the particulate organic pool during exponential growth. On average, TFA is estimated to represent 23% of POC for the three balloons (Figure 4).

### 3.3. Bulk POC and DIC and their ^13^C-Labeling

Upon addition of ^13^C-CO_2_ to the culture at t_0_, the atomic enrichment of the DIC stayed close to 0 till t_4_. After t_4_ and until t_24_, AE_DIC_ increased up to 60%, with a plateau after t_20_ (Figure 5). The DIC ^13^C level was similar between balloons (Figure 5). As for DIC, AE of POC remained close to zero until t_4_ and then sharply increased in both balloons (Figure 5). As both AE_POC_ and AE_DIC_ increased at t_4_–t_6,_ the ^13^C enrichment appeared thus to be almost simultaneous between DIC and POC pools. The level of enrichment of POC was higher in Cm1 than in Cm2 but remained parallel during the course of the experiment.

### 3.4. Fatty Acid Composition of Neutral and Polar Lipids in C. Muelleri

Saturated fatty acids (SFA) and monounsaturated fatty acids (MUFA) accounted for, respectively, 41% and 36% of neutral lipid fatty acids (on average for the three balloons), while PUFA represented 22% (Figure 6). Polyunsaturated fatty acids (PUFA) were more abundant in polar lipids (40% of total polar lipid fatty acids on average), while SFA and MUFA were at 33% and 26% of total polar lipid fatty acids, respectively. For both polar and neutral lipids, the main FA were 14:0, 16:0, 16:1n-7 and 20:5n-3 (Figure 6).

Fatty acids involved in the C_16_ PUFAs pathway (16:2n-7, 16:2n-4, 16:3n-4, 16:4n-1) and n-6 pathway (18:2n-6, 18:3n-6, 20:2n-6, 20:3n-6, 20:4n-6, 22:4n-6, 22:5n-6) represented, respectively, 8% and 7% of FA in the polar lipids (PL) versus 5% and 3% in the neutral lipids (NL). The proportions of FA involved in the n-3 pathway (18:3n-3, 18:4n-3, 20:4n-3, 20:5n-3, 22:5n-3 and 22:6n-3) represented 25% of FA in PL and 13% in NL. Appendix A gives the concentration in µg·L^−1^ and µmolC·L^−1^ for all fatty acids identified in this study.

### 3.5. Variability of Fatty Acid Proportions during the Experiment

The PERMANOVA analysis on NL and PL fatty acids revealed a significant difference between sampling times for fatty acid mass percentage (*p* < 0.001) but no significant difference between the three balloons (*p* > 0.05).

The PCA of neutral and fatty acids was coupled with SIMPER and revealed a clear time-dependent division for both NL and PL (Figure 7). The first sampling points were on the positive side, and the last sampling points were located on the negative side of the axis 1. In PL, axis 1 was driven by fatty acids 20:5n-3, 16:2n-7 and 16:3n-4 and in NL by 18:3n-6, 16:0, 16:1n-7 and 15:0. Following these multivariate analysis, 16:0 and 16:1n-7 were found to be correlated (Spearman test: α = 0.65 *p*-value < 0.05 in PL, α = 0.90 *p*-value < 0.05 in NL), as were 16:3n-4 and 20:5n-3 (Spearman test: α = 0.86 *p*-value < 0.05 in PL, α = 0.88 *p*-value < 0.05 in NL), 18:4n-3 and 20:5n-3 (Spearman test: α = 0.72 *p*-value < 0.05 in PL, α = 0.70 *p*-value < 0.05 in NL) and 20:4n-6 and 18:3n-6 (Spearman test: α = 0.72 *p*-value < 0.05 in PL, α = 0.69 *p*-value < 0.05 in NL). 18:2n-6 and 18:3n-6 were significantly related in PL (Spearman test: α = 0.93 *p*-value < 0.05) but not in NL (Spearman test: α = −0.07 *p*-value > 0.05).

Concentration (µmolC·L^−1^) dynamics of 16:3n-4, 20:5n-3, 16:0 and 16:1n-7 are shown in Figure 8. Both 16:0 and 16:1n-7 presented steady increases of their concentrations in both NL and PL fractions. The concentration doubled for 16:0 with time, on average from 122 ± 5 µmolC·L^−1^ to 205 ± 9 µmolC·L^−1^ and 322 ± 34 µmolC·L^−1^ to 750 ± 10 µmolC·L^−1^, respectively, for PL and NL and almost doubled for 16:1n-7 from 209 ± 22 µmolC·L^−1^ to 327 ± 17 µmolC·L^−1^ and from 504 ± 62 µmolC·L^−1^ to 943 ± 60 µmolC·L^−1^, respectively, for PL and NL. On the contrary, 16:3n-4 and 20:5n-3 concentrations dynamics oscillated with time. Decreases in concentrations of 16:3n-4 and 20:5n-3 for the NL fraction were concomitant with their opposite increases in the PL fraction. Reverse observations were also true. This pattern was particularly observed at t_2_, t_4_, t_10_ and t_14_. These oscillations were to put in relation to the correlations observed for NL and PL for 16:3n-4 and 20:5n-3 (Figure 7). After t_18_, the concentrations of 16:3n-4 and 20:5n-3 became more time-stable and more abundant within the PL fraction: 26.7 ± 3.3 µmol·L^−1^ and 103.1 ± 9.3 µmol·L^−1^ for 16:3n-4, 209.9 ± 27.3 µmol·L^−1^ and 383.1 ± 27.7 µmol·L^−1^ for 20:5n-3, respectively, for NL and PL in average for the three replicates (Figure 8). To a lesser extent, the same dynamics were found for 16:2n-7, 16:2n-4 and 18:4n-3 in both NL and PL fractions (data not shown).

### 3.6. Fatty Acid ^13^C-Enrichment and Synthesis

The atomic enrichments (AE) of the 10 studied fatty acids were gathered according to hypothesized pathways (C_16_ PUFAs, n-3 PUFAs, and n-6 PUFAs pathways and post-FAS pathway) (Figure 9).

For the first group (SFA + 18:1n-9), 16:0 and 18:1n-9 were the most enriched FA in PL, followed by 18:0. In NL, 16:0 was the most enriched, followed by 18:0 and 18:1n-9, which had the lowest enrichment. For C_16_ fatty acids, 16:1n-7 incorporated more ^13^C than 16:3n-4 in both NL and PL. Enrichment in ^13^C for these two fatty acids was similar between fractions. ^13^C levels of enrichment of 16:0 and 16:1n-7 were similar in NL, while 16:0 incorporated more ^13^C than 16:1n-7 in PL.

For n-3 PUFAs, 18:4n-3 was always more enriched in ^13^C than 20:5n-3 in both NL and PL that presented, like for C_16_ fatty acids, a relatively similar labeling. For the n-6 PUFAs, the ^13^C incorporation seemed to follow the carbon chain size, first the C_18_ PUFAs (18:2n-6 and 18:3n-6) and then 20:4n-6. For this group, PL was always more enriched than NL. By comparing PUFAs of the n-3 and n-6 together, the levels of ^13^C in polar lipids followed the order: 18:2n-6, 18:3n-6, 18:4n-3, 20:4n-6, 20:5n-3. It was different for neutral lipids: 18:3n-6, 18:4n-3, 18:2n-6, 20:4n-6, 20:5n-3. By considering all C_18_ fatty acids, 18:1n-9 was more enriched in ^13^C in PL than 18:2n-6 and 18:3n-6, while less enriched in ^13^C in NL than C_18_ PUFAs.

In PL, 16:0 (0.024 h^−1^), 18:1n-9 and 18:2n-6 (0.023 h^−1^) had the highest specific uptake rates, followed by 18:3n-6 (0.021 h^−1^) (Figure 10). For NL, the specific uptake rate remained below 0.02 h^−1^ for all fatty acids except for 16:0. 18:4n-3 and 18:3n-6 presented higher rates (0.015 h^−1^). The specific uptake rate of 18:0 was higher in NL in comparison with PL (0.013 h^−1^ versus 0.005 h^−1^). The specific uptake rate of 18:4n-3 was similar in NL and PL fractions. In the n-6 PUFAs, 18:3n-6 and 18:2n-6 had a higher µ_FA_ in the polar fraction than in the neutral fraction. The specific uptake rate of 18:1n-9 was also higher in PL than in NL (Figure 10).

Mean ratio of atomic enrichment (AE) for pairs of FA were calculated for neutral lipids (NL) and polar lipids (PL) to explore the precursor–product relationship between FA (Figure 10).

For the C_16_ pathway and connections between n-3 and n-6 pathways, all the ratios are below 1 for NL and PL (Table 4). It was also the case for 20:5n-3/18:4n-3 in PL and NL in the n-3 pathway, 20:4n-6/18:3n-6 in PL and NL and 18:3n-6/18:2n-6 in PL for the n-6 pathway. However in NL, 18:3n-6/18:2n-6 and 18:2n-6/18:1n-9 are above 1 with respective mean values at 1.25 and 2.14. Finally 18:2n-6/18:1n-9 is close to 1 for PL. The 18:0 can allow the slow production of 18:1n-9 in NL (ratios below 1) but 18:1n-9/18:0 ratio is above 1 in PL (Table 4). Figure 11 summarised the interpretation of the results obtained with ratio calculations.

## 4. Discussion

### 4.1. Growth and ^13^C Incorporation

This study aimed at investigating the synthesis pathways of 20:5n-3 in *C. muelleri* using ^13^CO_2_ labeling. The addition of ^13^CO_2_ did not disturb the algae physiology: growth, cell size, cell complexity, chlorophyll content. Cell viability remained high throughout the experiment as revealed by constant proportions of alive (SYTOX) and active (FDA) cells. This supported the good physiological state of the algae, which was able to produce FA (around 23% of POC). Algae growth was associated with an increase of POC and TFA concentrations. The ^13^C enrichments into DIC and POC were significantly detected after t_4_. After a lag phase, both DIC and POC pools were rapidly enriched, with slightly different kinetics. DIC enrichment increased rapidly at first and then stabilized after 14–16 h, while POC enrichment increased steadily following a linear slope. The difference of POC enrichment in ^13^C between the two labeled balloons remained unclear, especially since the respective levels of ^13^C found in the studied FA for both replicates stayed close. The incorporation of the ^13^C into fatty acids was detectable after 4 h, but their levels of enrichment varied according to fatty acids. ^13^C incorporation in some FA (16:0, 18:1n-9) was even higher than in POC.

### 4.2. The C_16_ PUFAs Pathway in C. muelleri

*C. muelleri* showed similar fatty acids profile as other diatoms: 14:0, 16:0, 16:1n-7 and 20:5n-3 (EPA) were the major fatty acids for polar and neutral lipids with PUFA higher in polar lipids and SFA and MUFA predominant in neutral lipids [61,62,63,64,65]. The strong concentration of 16:1n-7 and its high enrichment supported its central role as a precursor of the C_16_ PUFAs pathway. The C_16_ PUFAs pathway is initiated by 16:1n-7 which is first desaturated in 16:2n-7 or 16:2n-4 [66,67]. Then, 16:2n-4 is desaturated again in 16:3n-4 and finally in 16:4n-1. In our experiment, 16:4n-1 was only present in trace amounts, 16:2n-7 and 16:2n-4 were in low concentrations, and 16:3n-4 was the most concentrated. We assumed that the C_16_ PUFAs synthesis pathway led to the accumulation of 16:3n-4 at the expense of its precursors. The 16:1n-7 being much more enriched than 16:3n-4 confirmed that 16:1n-7 can be its precursor. The low concentration of 16:2n-7 and 16:2n-4 suggested that these intermediates in the C_16_ PUFAs pathway were rapidly converted toward 16:3n-4. Furthermore, as ^13^C accumulation or incorporation could not be detected within these two synthesis intermediates, we can hypothesize that the desaturation of 16:1n-7 (16:1Δ9) to 16:2n-7 (16:2Δ6, 9) or to 16:2n-4 (16:2Δ9, 12) was the limiting step in the C_16_ PUFAs pathway toward 16:3n-4 (16:3Δ6, 9, 12). The fact that 16:1n-7 was among the most enriched FA in both NL and PL might reflect some storage process in “anticipation” of later C_16_ PUFAs synthesis, energy shortage or stress response.

Some studies have focused on the role played by 16:3n-4 in microalgae and plants. 16:3n-4, formed in the chloroplast, has been found in galactolipids and more precisely in monogalactodiacylglycerol (MGDG) of diatoms like *Skeletonema costatum* and *Thalassiosira rotula* [68,69,70]. With 16:4n-1, 16:3n-4 was speculated to help algae to cope with environmental stress [70]. Indeed, 16:3n-4 is involved in aldehyde synthesis: by oxidation, small aldehydes are formed from polyunsaturated fatty acids and contribute to repairing or controlling damages [68,69]. Desbois et al. [71] showed that 16:3n-4 and 16:1n-7 of the diatom *Phaeodactylum tricornutum* also presented antibacterial proprieties. 16:3n-4 is able to reduce or kill Gram + and Gram − bacteria.

### 4.3. EPA Synthesis Resulting from a Combination of n-3 and n-6 Pathways?

The sum of n-3 fatty acids was higher than those of n-6 fatty acids in *C. muelleri*. As mentioned before, 20:5n-3 and 18:4n-3 were the most concentrated n-3 fatty acids. Based on this, it could be assumed that the diatom synthetized 20:5n-3 mainly through the n-3 pathway rather than through the n-6 pathway. However, 18:3n-3 and 20:4n-3, the respective precursors of 18:4n-3 and 20:5n-3 [14], were only found in low quantities. Then, hypothesizing synthesis pathways based on fatty acid abundance could be biased if the end product is highly concentrated. Furthermore, the alternative Δ8-desaturase pathway converting 18:3n-3 into 20:3n-3 and next into 20:4n-3 [17,72] was probably limited in *C. muelleri* because these two FA were only found in trace amounts. It seems more likely that 18:4n-3 and 20:5n-3 were produced via ω3-desaturation of 18:3n-6 and 20:4n-6, respectively [66,73]. Among n-6 fatty acids, 18:3n-6 and 20:4n-6 were the two most important n-6 PUFAs. The existence of the connection between n-3 and n-6 pathways had been previously demonstrated. Indeed, for the diatom *Phaeodactylum tricornutum*, Arao and Yamada [14] showed, using ^14^C labeled substrate, that 20:5n-3 was synthesized through both n-3 and n-6 pathways. The 18:2n-6 and 18:3n-6 can respectively be used to form the 18:3n-3 and 18:4n-3 by a ω3-desaturase [19,66]. 18:3n-6 can also be produced from 18:2n-6 by a Δ6-desaturase [19,66] (Figure 1).

In our experiment, the specific uptake rates of n-6 PUFAs seemed to follow the known n-6 pathway fatty acids for polar lipids (18:1n-9 → 18:2n-6 → 18:3n-6 → 20:4n-6) and the ω3-desaturase pathway (18:3n-6 → 18:4n-3 and 20:4n-6 → 20:5n-3). Similar observations were made for the neutral lipid fraction. These statements were also supported by the atomic enrichment ratios linking fatty acid precursors and products. This means that synthesis of 18:3n-6 from 18:2n-6, 18:4n-3 from 18:3n-6, or 20:4n-6 from 18:3n-6, as well as 20:5n-3 from 18:4n-3 and 20:4n-6, were possible in *C. muelleri*. Concerning 20:5n-3 synthesis, it remains, however, difficult to precisely assess the respective contribution of 18:4n-3 and 20:4n-6 to its production. We can only assume that both were likely to occur in *C. muelleri*.

### 4.4. Interconnection between Polar and Neutral Pools in C. muelleri

The successions of fatty acid enrichments differed between NL and PL. While enrichments of 18:1n-9, 18:2n-6, 18:3n-6 and then 20:4n-6 were in agreement with the conventional n-6 pathway used by *C. muelleri* to synthesize its PUFAs in the polar fraction [14,15,16,19], the order varied for neutral lipids. Furthermore, the higher enrichments of 18:1n-9, 18:2n-6 and 18:3n-6 in PL as compared to NL strongly suggested that their synthesis and more specifically the insertion of the new double bound (desaturation) occurred in the form of glycolipid or phospholipid and are transferred to neutral lipids once synthesized. This can be linked to triacylglycerol (TAG) synthesis.

In diatoms, TAG are synthesized in the endoplasmic reticulum (ER) from diacylglycerol (DAG) [17,32]. They are used as energy or carbon storage. The synthesis of TAG occurs through the acyl-CoA pathway, called the Kennedy pathway. Some authors also hypothesized the possibility of TAG synthesis through an acyl-CoA-independent pathway [17,34]. The de novo Kennedy pathway consists in three consecutive acylations: (i) acylation of the *sn*-1 position of G3P by G3P acyltransferase (GPAT), producing lysophosphatidic acid (LPA), (ii) following esterification of LPA *sn*-2 position by lysophosphatidic acid acyltransferase (LPAAT) to give PA and (iii) finally, after release of the phosphate group, DAG *sn*-3 position acylation by DAG acyltransferase (DGAT) to produce TAG (Figure 12).

In the acyl-CoA-independent pathway, the phospholipid:DAG acyltransferase (PDAT) needs PC as an acyl donor to form DAG. This pathway relies on the Lands cycle, which consists of an acyl-editing mechanism that progressively changes the fatty acids composition of PC to a more unsaturated one [33,37] and thus modifies the final composition of the newly formed DAG (and consequently TAG) (Figure 12). The enzyme responsible for the combination of PC and DAG, the phospholipid:diacylglycerol acyltransferase (PDAT), has been identified in plants [74,75]. Enzymes involved in this Lands cycle pathway linking PC with DAG to form TAG and genes coding for PDAT have been identified in the diatom *Phaeodactylum tricornutum* [35,36]. This could support the existence of this pathway in diatoms as well.

If we apply these putative pathways to the observed enrichment dynamics, it seemed likely that 18:1n-9 (PL) was first synthesized in the chloroplast or in the cytosol where its suspected precursors can be found and entered PC after being transferred to the ER to be rapidly desaturated in 18:2n-6 (PL) and 18:3n-6 (PL) by the acyl-editing machinery. Then, these newly formed PUFAs can enter the acyl-CoA pool and be used for de novo synthesis by Kennedy pathway in the ER or C_18_ PUFAs can directly be exchanged with a low unsaturated FA in DAG to create a more highly unsaturated DAG, which was then used for TAG synthesis. Such interaction between phospholipids pool and DAG/TAG pool was suggested by Mus et al. [36] and Mülhroth et al. [35] following the discovery of gene coding for putative enzyme involved in this process. This would mainly concern 18:2 and 18:3 fatty acids [32]. Combined with our results, this supports the implication of PC-associated-PUFA in the TAG synthesis via DAG in diatoms (Figure 12).

18:1n-9 was enriched in the neutral fraction after 18:2n-6 and 18:3n-6. The ratio 18:2n-6/18:1n-9 was above 1 in NL, which indicates an unlikely synthesis of 18:2n-6 from 18:1n-9 within NL. We hypothesize that 18:1n-9 was synthesized in the cytosol from 18:0 (18:1n-9/18:0 < 1 in NL) at a slower rate than the transfer rates of 18:2n-6 and 18:3n-6 from the PL compartment to the neutral (storage) compartment. This would support the observation that the synthesis of 18:1n-9 can take place in both the plastid and the cytosol [76]. 18:1n-9 could then be introduced into DAG, and further, TAG by the Kennedy pathway. “Prokaryotic” configuration TAG with 18:0 or 18:1n-9 at position *sn*-1 or *sn*-3 has already been reported in diatoms [77,78,79].

### 4.5. C_16_ PUFAs and 20:5n-3: Linked on Glycerol Backbone in C. muelleri

Interconnections between polar and neutral lipids pools in diatoms were also revealed by the opposite trends observed for 20:5n-3 and 16:3n-4 in NL and PL fractions (concentrations of NL and PL fractions were anticorrelated: Spearman test, α = −0.66, *p*-value < 0.001 for 20:5n-3 and α = −0.73 for 16:3n-4, *p*-value < 0.001). These dynamics suggested some transfers between pools during *C. muelleri* growth. Moreover, both 16:3n-4 and 20:5n-3 were also correlated within the pool as shown with PCA analysis as well as with their respective atomic enrichment correlations (AE of 20:5n-3 and AE of 16:3n-4 were correlated: Spearman test α = 0.97, *p*-value < 0.001 and α = 0.99, *p*-value < 0.001, respectively, for NL and PL). However, these fatty acids were not directly associated in synthesis pathways (Figure 1). Consequently, these statistically connected dynamics would suggest 20:5n-3 and 16:3n-4 to be located on the same glycerol backbone and then presenting concomitant variation concentrations and enrichments between and/or within pools.

Based on cloning and functional characterization of *P. tricornutum* desaturases, Domergue et al. [66] and Mülhroth et al. [35] discussed the existence of connected pathways in diatoms between C_16_-PUFAs and 20:5n-3. Their idea came from the existence of a glycolipid presenting 16:3n-4 and 20:5n-3 at analogous positions (respectively position *sn*-2 and *sn*-1) of typical glycolipids (GL) produced via the prokaryotic pathway in *P. tricornutum*. Indeed, depending on the location of the synthesis of glycolipids (i.e., chloroplast “prokaryotic” or endoplasmic reticulum (ER) “eukaryotic”), the fatty acid composition varies: GL produced by the prokaryotic pathway will have C_16_ fatty acids at the *sn*-2 position, while those formed in the ER will present longer chain fatty acids (likely C_18_/C_20_ or C_22_) at *sn*-2 position [80,81]. Our results suggested the plastid origin of the glycolipids linking 20:5n-3 (*sn*-1) and 16:3n-4 (*sn*-2). However, 20:5n-3 synthesis and the responsible desaturase in *P. tricornutum* seemed exclusively located out of the chloroplast, contrasting with higher plant metabolism [66]. Then, to be added to chloroplast glycolipids, 20:5n-3 had to be transferred from the ER to the plastid. Transfers of FA precursors from the ER to the plastid have been reported to exist in plants and microalgae [17,82].

Thus, as for other diatoms and following the results introduced here, a structural and metabolic link could exist between 16:3n-4 and 20:5n-3 in *C. muelleri*, even if they were not synthesized in the same cell compartments (Figure 13). 20:5n-3 synthesized in the ER was assumed to be used to form prokaryotic glycolipids in association with 16:3n-4 produced in the chloroplast. Along the course of its synthesis, the molecule passed through polar and neutral pools, for example, PA to DAG (PL to NL pool) or DAG to glycolipids (NL to PL pool), which would explain the opposite dynamics observed in this experiment. Similar dynamics observed with 18:4n-3 could also be related to eukaryotic DAG synthesis in the ER (Figure 13).

## 5. Conclusions

*C. muelleri* synthetized 20:5n-3 via the combined use of n-3 and n-6 pathways. The conventional route via 18:3n-3 seemed inefficient and/or slow in the diatom: apart from 18:4n-3 and 20:5n-3, all the other n-3 PUFAs were present in low concentrations. As an alternative, *C. muelleri* appeared to produce its n-3 PUFAs from the n-6 substrate via the ω-3 desaturase pathway. This was especially the case of 18:3n-6, transformed into 18:4n-3 and 20:4n-6 converted into 20:5n-3. It was not possible to discriminate which PUFA between 20:4n-6 and 18:4n-3 was predominant in EPA synthesis. The use of ^13^C-label allowed us as well to assume the existence of an acyl-editing mechanism in *C. muelleri* responsible for the progressive desaturation of 18:1n-9. Initially produced in the plastid, this fatty acid would be transmitted to the ER combined with phosphatidylcholine (PC) to be desaturated into 18:2n-6 and 18:3n-6. This would allow the formation of a more unsaturated diacylglycerol, serving in turn to format of triacylglycerol or membrane lipids. *C. muelleri* relied on a tight connection between the neutral and polar pools. In the diatom, synthesis of galactolipids in the chloroplast might be dependent on acyl-CoA produced in the ER. This has been assumed with the opposite abundance dynamic of 16:3n-4 and 20:5n-3, which were both suspected to be located on the same glycerol backbone of glycolipids. 16:3n-4 would be produced in the plastid while 20:5n-3 by successive desaturation and elongation occurring in the ER.

## Figures and Tables

**Figure 1 biomolecules-10-00797-f001:**
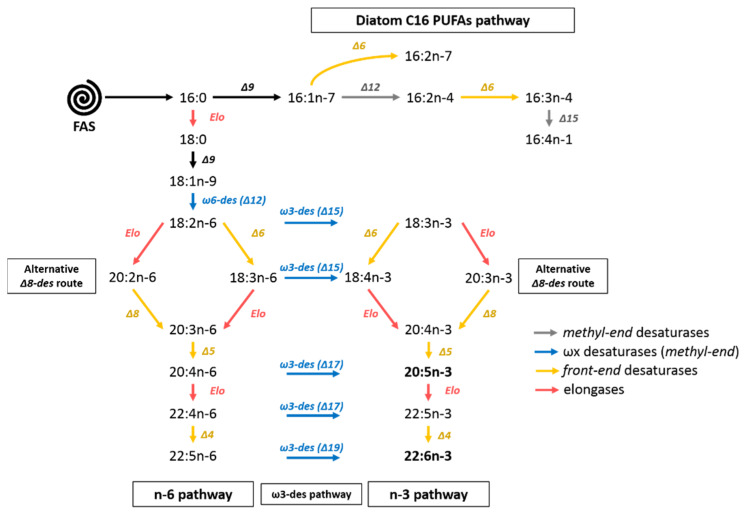
Fatty acids synthesis in diatoms. Desaturases are noted with “ω” and/or “Δ” and refer to the location of carbon holding the newly formed double bond and its position within the methyl or front end of the acyl chain (2 types, *methyl-end* (grey or blue arrows) or *front-end* desaturases (yellow arrows)). Des: desaturase, Elo: elongase, FAS: fatty acid synthase.

**Figure 2 biomolecules-10-00797-f002:**
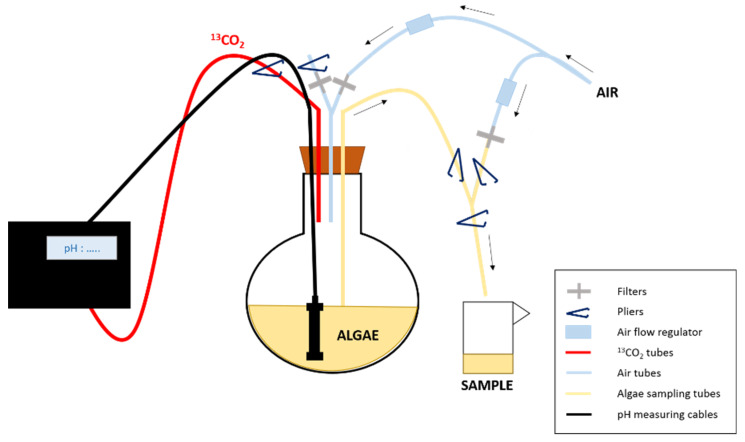
Experimental design of the enrichment experiment. The ^13^CO_2_ is supplied to the culture using a pH-stat system. To prevent contamination when sampling the algae, pliers are used to close/open the tubes/ways needed to first put the balloon under pressure and then allow sampling and finally rinse the tubes after collection.

**Figure 3 biomolecules-10-00797-f003:**
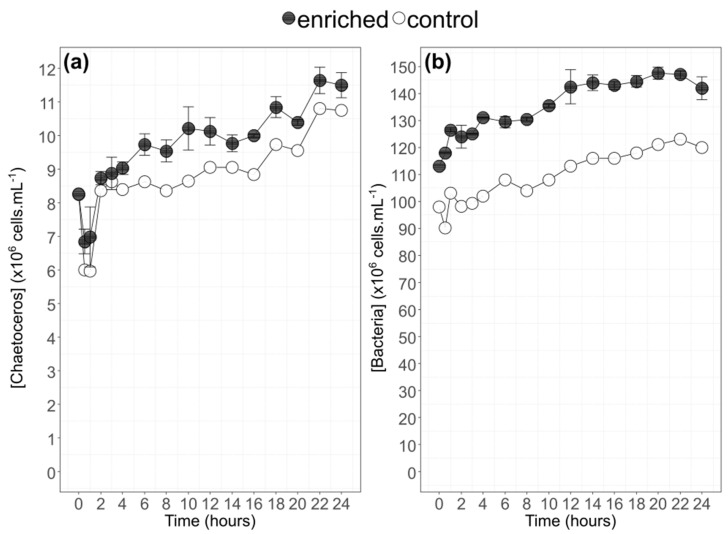
Dynamics of cell concentrations of the two enriched balloons of *Chaetoceros muelleri* (mean ± SD; solid line ● n = 2) and of the control balloon (solid line ○) (**a**) and the corresponding bacteria concentrations (**b**) during the 24 h experiment.

**Figure 4 biomolecules-10-00797-f004:**
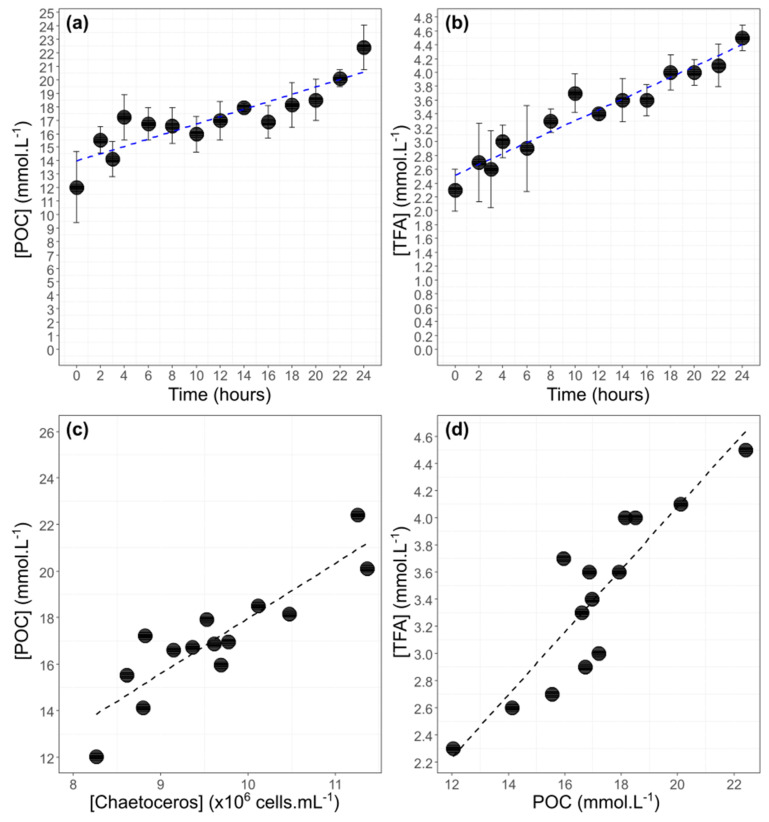
Dynamics of Particulate Organic Carbon concentration (**a**) and of Total Fatty Acids concentration (**b**) for the two enriched balloons Cm1 and Cm2 (Cm = *Chaetoceros muelleri*) (mean ± SD; n = 3). The blue line corresponds to increasing trend between t_0_ and t_24_ (POC: y = 0.27x + 14, R^2^ = 76% *p*-value < 0.001; TFA: y = 0.08x + 2.5, R^2^ = 93% *p*-value < 0.001). POC concentration relationships with algae concentration (**c**) (eq: y = 2.4x − 5.7 R^2^ = 79%, *p*-value < 0.001) and with total fatty acid concentration (**d**) (eq: y = 0.23x − 0.5, R^2^ = 80%, *p*-value < 0.001) (mean; n = 3). Points below 8 × 10^6^ cells·mL^−1^ are not included in the linear regression and are considered as outliers (corresponding to t_0.5_ and t_1_ in Figure 3).

**Figure 5 biomolecules-10-00797-f005:**
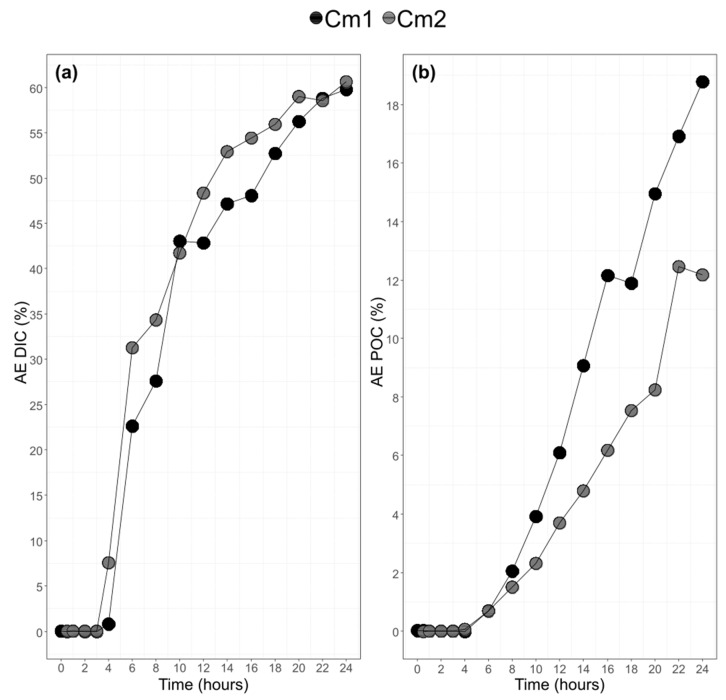
Atomic enrichment (AE) of the Dissolved Inorganic Carbon (**a**). Atomic enrichment (AE) of Particulate Organic Carbon (**b**) (Cm = *Chaetoceros muelleri*).

**Figure 6 biomolecules-10-00797-f006:**
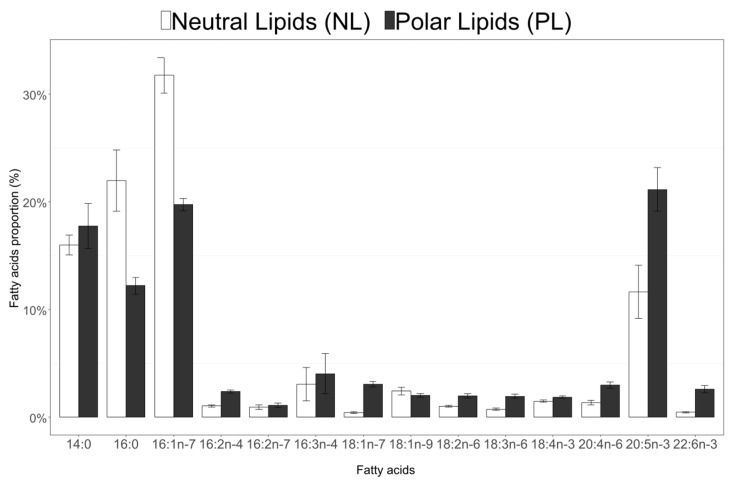
Proportions (% of Total Fatty Acids) of main fatty acids in neutral lipids (NL) and polar lipids (PL) of *C. muelleri*. (Mean ± SD n = 16 for each fraction). Among the 44 identified fatty acids (FA), only those accounting for more than 1% of the TFA are represented here (between 0.1% and 1%, 15:0, 18:0, 22:0, 14:1n-5, 16:1n-5, 22:2n-6 were detected in both neutral and polar lipids, iso15:0, 24:0, 16:1n-9, 17:1n-7, 16:2n-6 and 16:4n-3 were detected in neutral lipids and 18:1n-11, 18:1n-5, 16:4n-1, 18:3n-4, 20:3n-6, 20:4n-3 and 22:5n-3 were detected in polar lipids). The proportions of the 10 presented FA were significantly different between neutral lipids and polar lipids.

**Figure 7 biomolecules-10-00797-f007:**
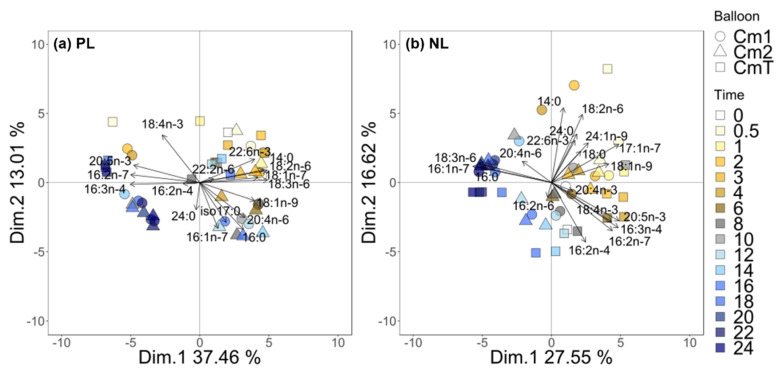
Principal component analyses (PCA) made with %mass for all the fatty acids in the polar fraction (PL) (**a**) and the neutral fraction (NL) (**b**). Only the FA selected by SIMPER analysis are shown here (explaining 80% of the variability). The colors represent the different sampling times from t_0_ to t_24_, and the symbols represent the balloons considered. Cm1 and Cm2 are the enriched balloons, CmT the control balloon (Cm = *Chaetoceros muelleri*, n = 48 for NL and n = 48 for PL).

**Figure 8 biomolecules-10-00797-f008:**
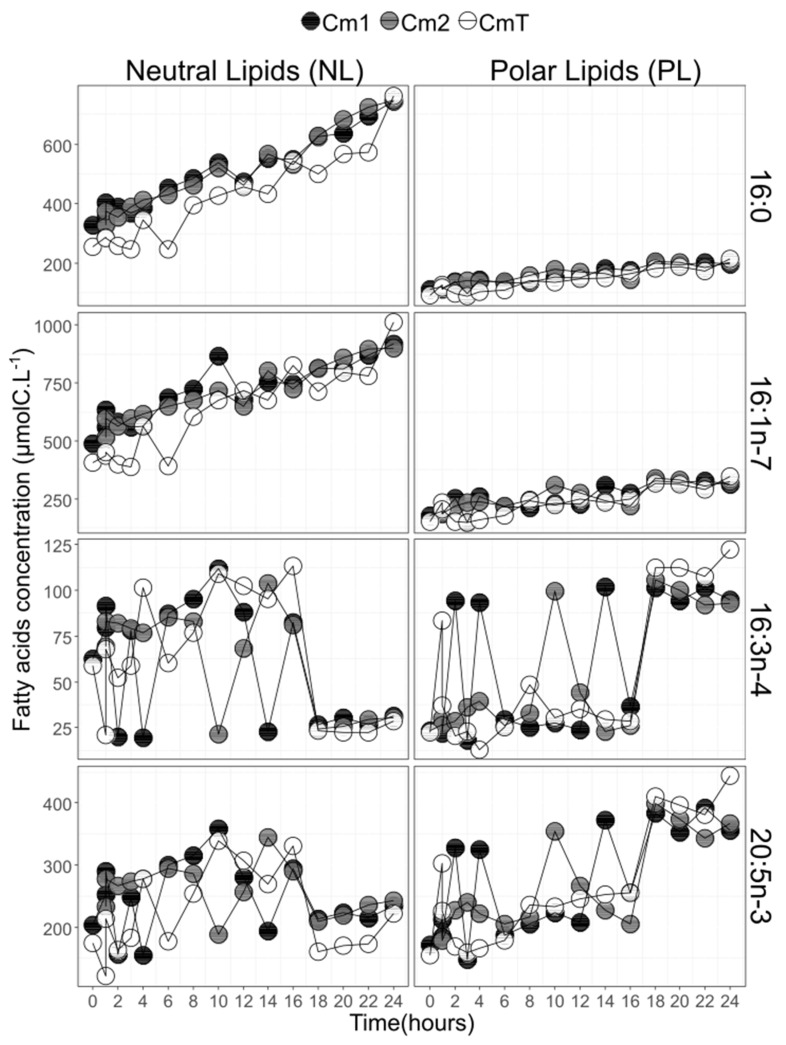
FA concentration dynamics in neutral lipid fraction (NL—left) and polar lipid fraction (PL—right) with time for the two most abundant C_16_ fatty acids (16:0 and 16:1n-7), and the two end-product fatty acids (16:3n-4 and 20:5n-3) for the three culture balloons. Cm1 and Cm2 are the enriched balloons, and CmT is the control balloon (Cm = *Chaetoceros muelleri*)**.**

**Figure 9 biomolecules-10-00797-f009:**
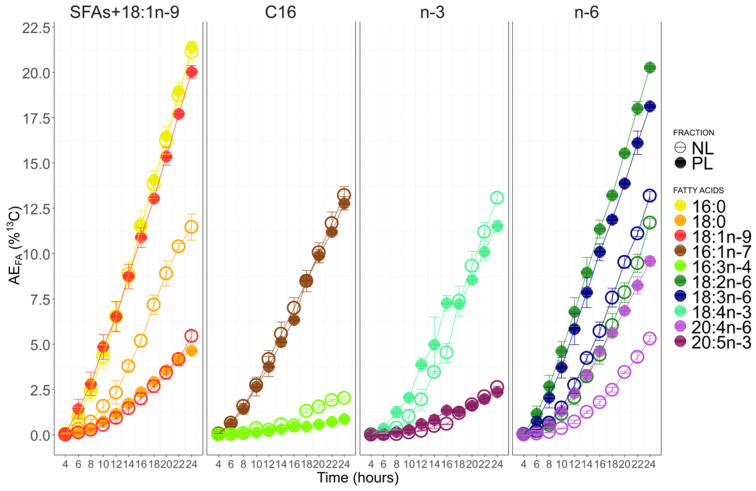
Atomic enrichment of the 10 most abundant FA in polar lipids (PL) and neutral lipids (NL) and suspected as involved in PUFA synthesis in *C. muelleri*. The standard deviation represents the variability between enriched balloons (Cm1 and Cm2, Cm = *Chaetoceros muelleri*).

**Figure 10 biomolecules-10-00797-f010:**
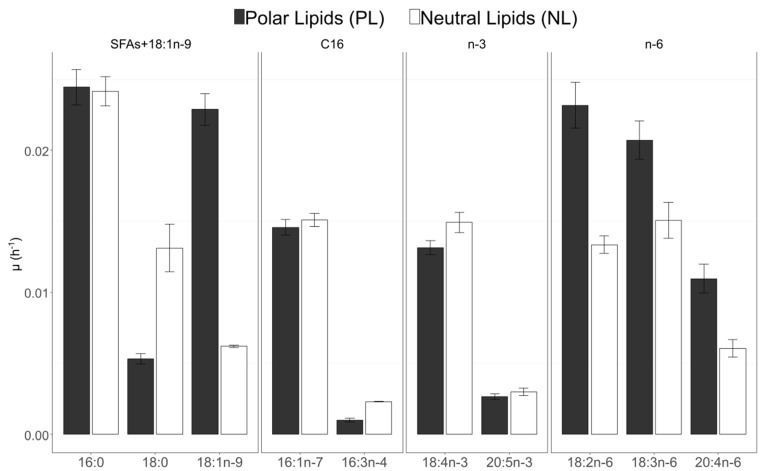
Specific uptake rates calculated with AE (µ_FA_) of the 10 studied fatty acids in polar lipids (PL) and neutral lipids (NL) of *C. muelleri* (n = 2 for each).

**Figure 11 biomolecules-10-00797-f011:**
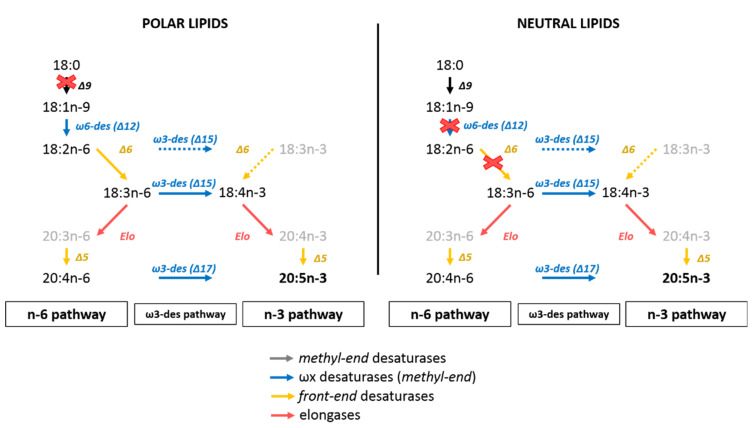
Summary of FA synthesis relationships in polar and neutral lipids pools for *C. muelleri* according to ratio calculation results. The solid arrows correspond to ratios of fatty acid B and suspected precursor A that are equal to or below 1, and red crosses correspond to ratios above 1. The fatty acids in grey were absent or present in trace amounts in this experiment, and consequently, the ways marked by dashed arrows were not possible to verify.

**Figure 12 biomolecules-10-00797-f012:**
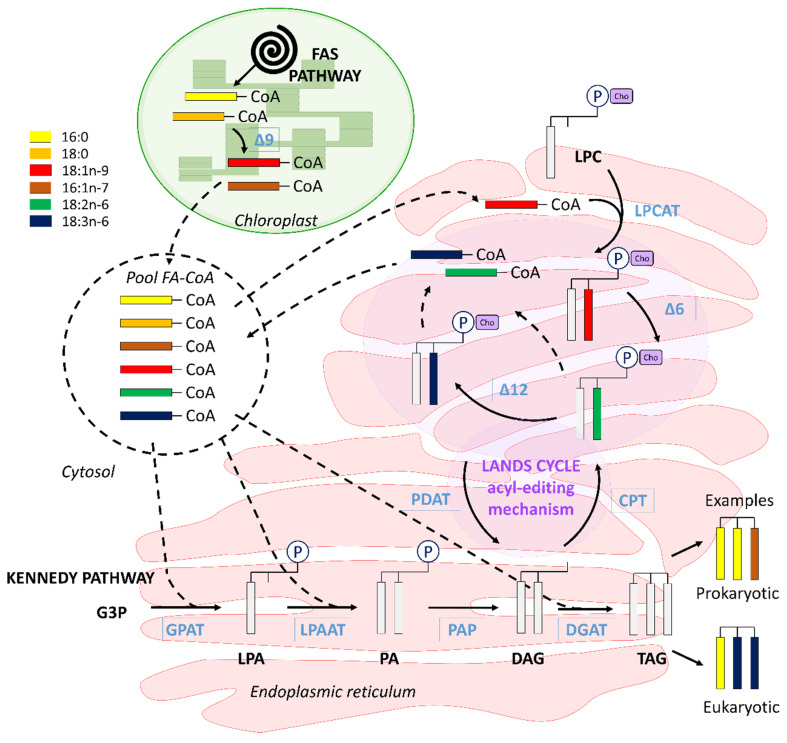
Hypothesized synthesis of phosphatidylcholine (PC) and triacylglycerol (TAG) in *C. muelleri*. CoA: co-enzyme A, CPT: cytidine-5′-diphosphate choline phosphotransferase, DAG: diacylglycerol, DGAT: acyl-CoA DAG acyltransferase, G3P: glycerol-3-phosphate, GPAT: acyl-CoA-glycerol-3-phosphate acyltransferase, LPA: lysophosphatidic acid, LPC: lysophosphatidylcholine, LPAAT: acyl-CoA-lysophosphatidic-acid- acyltransferase, LPCAT: acyl-CoA LPC acyltransferase, PA: phosphatidic acid, PAP: acyl-CoA-phosphatidic acid phosphatase, PC (P-Cho): phosphatidylcholine, PDAT: phospholipid:DAG acyltransferase, TAG: triacylglycerol. Δ9, Δ12, Δ6 desaturases.

**Figure 13 biomolecules-10-00797-f013:**
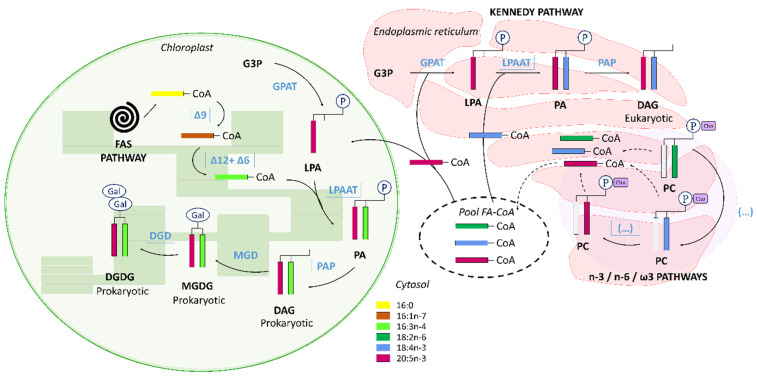
Hypothesized pathway for the production of prokaryotic glycolipids in *C. muelleri*. Two pathways can produce DAG: one is located in the chloroplast (*prokayotic pathway*), the other in the endoplasmic reticulum (ER) (*eukaryotic pathway*). The 20:5n-3-CoA used for glycolipids synthesis is produced in the ER and collected from it to form prokaryotic DAG. This DAG can be further used to create glycolipids (configuration *sn-1*: 20:5n-3 and *sn-2*: 16:3n-4). The eukaryotic pathway in the ER could also explain the similar concentration dynamics observed between 20:5n-3 and 18:4n-3. CoA: co-enzyme A. DAG: diacylglycerol, DGD: digalactodiacylglycerol synthase, DGDG: digalactodiacylglycerol, G3P: glycerol-3-phosphate, GPAT: acyl-CoA-glycerol-3-phosphate acyltransferase, LPA: lysophosphatidic acid, LPAAT: acyl-CoA-lysophosphatidic-acid-acyltransferase, MGD: monogalactodiacylglycerol synthase, MGDG: monogalactodiacylglycerol, PA: phosphatidic acid, PAP: acyl-CoA-phosphatidic acid phosphatase, PC: phosphatidylcholine.

**Table 1 biomolecules-10-00797-t001:** List of international and in-house standards used for EA-IRMS and GB-IRMS analysis. IAEA: International Atomic Energy Agency.

Description	Nature	Analysis	δ^13^C (‰)	SD
IAEA-CH_6_	Sucrose (C_12_H_22_O_11_)	^13^C-POC	−10.45	0.03
IAEA-600	Caffeine (C_8_H_10_N_4_O_2_)	^13^C-POC	−27.77	0.04
Acetanilide	Acetanilide (C_8_H_9_NO)	^13^C-POC	29.53	0.01
CA21 (in-house std)	Calcium carbonate (CaCO_3_)	^13^C-DIC	+1.47	
Na_2_CO_3_ (in-house std)	Sodium carbonate	^13^C-DIC	−6.88	
NaHCO_3_ (in-house std)	Sodium bicarbonate	^13^C-DIC	−5.93	

**Table 2 biomolecules-10-00797-t002:** Groups of fatty acids formed according to their main pathways.

Group	Fatty Acids Concerned
C_16_ PUFAs pathway	16:2n-7/16:2n-4/16:3n-4/16:4n-1
n-3 pathway	18:3n-3/18:4n-3/20:4n-3/20:5n-3/22:5n-3/22:6n-3
n-6 pathway	18:2n-6/18:3n-6/20:2n-6/20:3n-6/20:4n-6/22:2n-6/22:5n-6
FAS and post FAS	16:0/18:0/16:1n-7/18:1n-9
Bacterial FA	15:0/17:0/iso15:0/iso17:0/17:1n-7

**Table 3 biomolecules-10-00797-t003:** Cellular parameters of *C. muelleri* using flow cytometry analysis (Mean ± SD, n = 3) according to sampling time. Control and enriched balloons were combined for this table as no differences were observed between balloons. Values for FDA and SYTOX are in %, values for BODIPY/FL3/SSC/FSC in arbitrary unit (a.u).

	FL1-FDA ACTIVE (%)			FL1-SYTOX ALIVE (%)			FL1-BODIPY (a.u)			FL3 (a.u)			SSC (a.u)			FSC (a.u)		
0	93	±	4	99	±	1	160	±	20	263	±	2	48	±	1	149	±	0
0.5	90	±	4	99	±	1	215	±	23	274	±	20	50	±	2	152	±	1
1	93	±	3	99	±	0	233	±	31	276	±	18	56	±	10	151	±	2
2	93	±	2	100	±	0	209	±	11	275	±	19	57	±	10	153	±	1
3	93	±	2	100	±	0	245	±	25	263	±	3	53	±	2	152	±	1
4	92	±	1	98	±	2	248	±	26	260	±	2	53	±	1	152	±	2
6	92	±	2	98	±	2	194	±	17	260	±	2	50	±	2	152	±	1
8	90	±	1	98	±	3	248	±	17	260	±	2	53	±	2	153	±	3
10	90	±	1	98	±	2	157	±	2	262	±	3	52	±	3	153	±	1
12	89	±	4	99	±	1	163	±	26	262	±	4	51	±	2	154	±	0
14	89	±	2	97	±	4	276	±	24	261	±	4	53	±	0	156	±	1
16	86	±	3	97	±	3	284	±	9	261	±	4	52	±	2	155	±	1
18	87	±	2	98	±	3	163	±	11	259	±	6	52	±	3	156	±	3
20	88	±	4	99	±	2	308	±	9	258	±	6	56	±	1	155	±	1
22	87	±	4	96	±	5	269	±	16	258	±	5	55	±	2	154	±	1
24	88	±	3	99	±	2	239	±	5	257	±	6	54	±	1	154	±	2

**Table 4 biomolecules-10-00797-t004:** Mean ratio of atomic enrichment (AE) for pairs of FA (FA_A_ vs. FA_B_) in neutral lipids (NL) and polar lipids (PL) (Mean ± SD, n = 10 sampling dates) for the two enriched balloons (Cm1, Cm2; Cm = *Chaetoceros muelleri*). If the ratio is equal to or close to 1, A and B are assumed at equilibrium, and B is synthesized quickly from A; if the ratio is below 1, the transformation of B from A is possible but slow. Finally, if the ratio is above 1, A is not the main precursor of B, which has to be synthesized by a different pathway.

	Neutral Lipids				Polar Lipids			
	Cm1		Cm2		Cm1		Cm2	
Fatty acid B/Fatty acid A	Mean	SD	Mean	SD	Mean	SD	Mean	SD
18:1n-9/18:0	0.37	0.03	0.41	0.05	5.72	2.16	5.89	2.14
18:2n-6/18:1n-9	2.25	0.15	2.03	0.27	1.00	0.06	0.98	0.07
18:3n-6/18:2n-6	1.23	0.14	1.26	0.11	0.83	0.10	0.85	0.08
20:4n-6/18:3n-6	0.33	0.05	0.29	0.09	0.43	0.09	0.41	0.10
20:5n-3/18:4n-3	0.18	0.05	0.13	0.05	0.18	0.02	0.15	0.04
18:4n-3/18:3n-6	0.77	0.20	0.82	0.20	0.62	0.06	0.61	0.08
20:5n-3/20:4n-6	0.40	0.09	0.37	0.12	0.26	0.03	0.23	0.04
16:1n-7/16:0	0.63	0.03	0.63	0.01	0.58	0.03	0.57	0.03

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
