# Peer review of "Study of Synthesis Pathways of the Essential Polyunsaturated Fatty Acid 20:5n-3 in the Diatom Chaetoceros Muelleri Using 13C-Isotope Labeling"

_biomolecules, 2020, doi:10.3390/biom10050797_

Round 1
Reviewer 1 Report
Please see comments in attached document.

Reviewer 2 Report
the paper by Remize et al describes the characterisation of the microalgae Chaetoceros Mueller for the incorporation of 13C in the lipid metabolism, with particular attention to omega-3 production.
The subject is interesting and research on the subject is needed.
The critical point of this paper is the fatty acid identification. As shown in Table 2, there are 5 fatty acid groups to be recognised and the authors describe the use of two simultaneous injections of the same sample onto two different columns (one apolar and one polar), and this can be perfectly understood since the elution on the two columns are different. However, both columns are 30 m long, and the oven is not specified, therefore there are important doubts on the possibility that the 26 fatty acids described in Table 2 PLUS the C23:0 reference, can be satisfactory separated.
The elution onto the two columns must be carried out first of all showing that the appropriate standards are separated, and this is also useful to give quantitative data.
Therefore, this referee requires to add a Supplementary Part with all the GC analyses performed, showing the good separation, the calibration procedures, the quantitation used for the analyses. Without this Supplementary part the data presented in the paper cannot be considered.
Another requirement for calibration of PUFA is to use one experiment of transesterification under the conditions used (H2SO4/MeOH 100 °C for 10 min) using a known quantity of EPA-containing triglyceride and showing that there is no consumption of EPA for heating at gin temperature under acidic condition. In fact, the use of saturated fatty acid C23:0 as standard is not sufficient to claim that the quantitation of PUFA is correctly done in the analyses.
Minor Points:
1) The authors show in Figure 1 the fatty acid metabolism of interest to the paper. However, they use wrong acronyms on the arrows regarding the elongation pathways, since these pathways cannot be recognised by the delta-x notation. The Elo pathways must contain only the Elo notation.
2) in the paper there are a lot of "error! Reference source not found" notations, that I cannot understand where it come from.
Round 2
Reviewer 1 Report
The manuscript is improved with the corrections. My main comment at this point is that while the supplementary data (i.e. chromatograms and table) is helpful, it needs some connection to the main body of the text. In other words, somewhere in the text the authors need make mention that such supplementary data is available.
And, within the file with the chromatograms itself, the authors should provide some sort of description, with figure legends, like they did in their response to the other reviewer. Otherwise, the file is just a collection of chromatograms and the reader probably won't know why they're included.
As a minor comment, in the Refernces section, the authors should:
- Italicize organism names.
- Make sure the first letter of a genus name is capitalized.
- Make sure that the number value in an isotopic label like 13C is superscripted.
- Make sure article titles all have the same format with regard to capitalization.
Reviewer 2 Report
The authors gave the answer to my question regarding the GC analysis and provided the requested GC traces. They told that they are expert in the analysis of the diatom lipids, but sometime to be expert in a field is not equivalent to be the only referee of their procedure.
As far as this referee is concerned, based on my personal expertise of the lipid analysis I still believe that the method of 30m GC column and the separation achieved among the fatty acids is not appropriate for the complexity of the sample. The authors should provide at least one GC trace using the DB23 column of 60 or 100 m length also because they did not provide any other proof of the double bond positions, and the DB23 is able to separate fatty acids in a more progressive manner (according to the carbon chain length and number of double bonds).
I firmly believe that for an analytical laboratory it is necessary to work on two different separation methods, to be safe enough for the recognition of the fatty acids. In alternative the GC/MS analysis and detailed fragment examination must be performed.
I see also that the authors use abbreviations that are really not usual in fatty acid nomenclature:
line 337: TFA (total fatty acids) that normally TFA are trans fatty acids
line 370 SATURATED FATTY ACIDS (SAFA) instead of SFA
in Figure 11 they use the sequence "n-3 pathway ω-3 pathway n-6 pathway" which is really not usual in fatty acid biosynthesis, since n-3 and ω-3 are the same thing.
